# The Impact of E-Learning Technologies on Entrepreneurial and Sustainability Performance

**Sichu Liu, Hongyi Sun \*, Jiahao Zhuang**  **and Rui Xiong** 

Department of Systems Engineering, City University of Hong Kong, Hong Kong, China;
sichuliu2-c@my.cityu.edu.hk (S.L.); zhuangjiahao777@gmail.com (J.Z.); ruixiong2-c@my.cityu.edu.hk (R.X.)
\* Correspondence: sun.3333@cityu.edu.hk

**Abstract:** After the pandemic, education will not go back to a 100% offline mode since the application of e-learning technologies (ELTs) cannot be avoided. Therefore, their impact should be studied for future education development. Most future entrepreneurs are attending school today. Therefore, universities need to supply necessary education to encourage students to cope with future conditions and development. At the same time, due to the increasing attention being paid to the harmony between ecology and prosperity, the sustainability aspect of entrepreneurship education needs to be emphasized as well. This study investigates the impact of ELTs on entrepreneurial education performance (including personal skills, product skills, and business skills), sustainability efficacy, and their impact on sustainability awareness. Data were collected from a master's degree class on entrepreneurship at a Hong Kong university and SmartPLS was used to analyze the data. It was found that ELTs have a significant relationship with entrepreneurial performance and sustainability efficacy. Meanwhile, sustainability efficacy also has a significant relationship with sustainability awareness. However, no significant relationship between entrepreneurial skills and sustainability awareness was identified. The results indicate that ELTs can improve students' entrepreneurial skills and sustainability awareness, which proves the effectiveness of ELTs and provides support for their application in future entrepreneurship education.

**Keywords:** e-learning technologies (ELTs); entrepreneurship education; sustainable efficacy; sustainability awareness



## 1. Introduction

### 1.1. E-Learning and Entrepreneurship Education

Technology has the ability to change the ways in which knowledge is acquired in human society, and to restructure traditional teaching and learning models [1]. As information and communication technology (ICT) are growing more sophisticated and convenient, they are increasingly penetrating the teaching environment of education, which makes them one of the alternative teaching environments besides traditional face-to-face teaching. Influenced by COVID-19, in the field of education, the transition to an online education mode is particularly fast [2]. The pandemic has stimulated the application of ELTs, which has brought changes to the teaching process in universities and has expanded the channels of interaction between teachers and students. During this period, many universities shifted from offline teaching to online teaching [3]. Universities around the world have adopted online learning since then [4].

However, for many academic institutions, this transformation was forced. At that time, there was no choice but to change traditional educational methods and completely shift to online teaching [5]. Onsite teaching was considered good, while other teaching methods aimed at coping with the pandemic and were simply a temporary approach [6]. This approach to accessing an education ecosystem that is urgently created in a crisis is called "Emergency Remote Learning" (ERT) [7]. In the post-pandemic era, most higher education

institutions in Hong Kong are eagerly resuming offline teaching, and ELTs are gradually becoming marginalized. Some teachers have even cancelled online classes. We support that the application of ELTs in higher education and academic institutions should be promoted instead of being abandoned. This paper aims to use entrepreneurship education as an example to illustrate the effectiveness of ELTs, which may change the views of teachers and students on this teaching method [8].

The field of entrepreneurship has received increasing academic attention worldwide [9]. Universities offer entrepreneurship education programs to cultivate entrepreneurial talents and provide opportunities for new startups [10].

### 1.2. Sustainability and Entrepreneurship Education

Sustainable development has become an increasingly popular topic, one of the reasons being that it is a balance and harmony between the economic, social, and environmental aspects, which can provide benefits for stakeholders in multiple different fields [11]. Starik pointed out that when they were first introduced, environmental and social initiatives were considered to add to the legal and moral burden of businesses [12]. However, nowadays, entrepreneurship has been recognized as one of the main channels for achieving sustainable development in all aspects. Creating new businesses is seen as a solution to social and environmental issues [13]. Therefore, the United Nations (UN) is increasingly encouraging and supporting ambitious young people in starting new businesses and creating more employment for themselves and others [14].

Peloza's research has demonstrated a positive relationship between corporate social responsibility and financial performance [15]. There are many similar studies that have made sustainability increasingly important in the entrepreneurial process [16]. On the contrary, little research has confirmed how sustainability in entrepreneurship education affects the entrepreneurial philosophy of future entrepreneurs. The integration of sustainability-related themes with entrepreneurship education is very low, although it was already proposed in the 1960s [17]. There has been a lack of support for sustainability-related educational subjects in business courses [18]. This study will explore the level of sustainability awareness by conducting a questionnaire asking questions related to sustainable development among students who participated in entrepreneurship courses. The objective is to explore appropriate teaching methods that combine sustainability and entrepreneurship in university courses and to add sustainability education elements to entrepreneurship education.

### 1.3. Structure of the Paper

This paper uses a survey method to study the impact of ELTs on entrepreneurship education performance and sustainability efficacy among students at a Hong Kong university after the pandemic. The objectives include:

1. The impact of ELTs on students' learning in terms of personal skills.
2. The impact of ELTs on students' learning in terms of product skills.
3. The impact of ELTs on students' learning in terms of business skills.
4. The impact of ELTs on students' sustainability efficacy.
5. The impact of entrepreneurial skills and sustainability efficacy on students' sustainable awareness.
6. Suggestions for entrepreneurial and sustainable education in the future.

## 2. Literature Review and Model Development

### 2.1. Definition and Concepts

In a survey of 200 presidents from the top 1000 universities conducted by Times Higher Education, "19 percent think that digital technology will have eradicated physical lectures by 2030, compared with 65 percent who disagree" [19]. Another 2018 study also showed that European presidents of top universities agree that digital technology will eradicate in-person lectures more widely than their American colleagues but to a lesser

extent than their Asian colleagues [20]. These two surveys indicate that the media and modes of communication in education have changed. E-learning is more widely used in education, even to some extent compensating for the shortcomings of in-person lectures.

Multiple terms often vary depending on the expertise and interests of researchers, making it difficult to find a universally accepted definition for the term e-learning [21]—e-learning, distance education, online learning, web-based education, and other names are all concepts that have previously been used in the literature [22]. Nonetheless, the definition of e-learning in most studies has commonalities, in that e-learning is generalized as a teaching model supported by information technology, where the technology used is referred to as e-learning technology (ELT) [23]. This can be understood better when the concept is placed in a learning environment [24], such as entrepreneurship education studied in this paper.

The predecessors of e-learning go back to the 19th century, when teachers began to use mail and shorthand technology for teaching activities [25]. At that time, it was called distance education, and the word e-learning officially appeared in the education field in the mid-1990s [26]. E-learning is a special teaching system. The basic elements of this system are computers and the internet, which enables the system to be in motion [27].

In the post-pandemic era, the learning environment is undergoing changes. We are transitioning from a single teaching method to a multi-channel learning approach. Higher education institutions should learn from and analyze interesting data from the development of educational institutions in the past few years, rethink and update their teaching models to benefit from technological changes, and explore the potential of new teaching models through various technological means to improve teaching efficiency [6]. Therefore, if technology is incorporated into effective teaching strategies, it can contribute to improving traditional learning procedures without limiting learning practices [28].

An entrepreneur's profile is crucial for the success of their entrepreneurial behavior, but some studies show that developing entrepreneurial characteristics with the help of educational institutions is equally important [29]. These entrepreneurial characteristics are also known as entrepreneurship skills, which refers to the abilities that entrepreneurs may possess to run a business [30,31]. Education plays a crucial role in forming them. From some perspectives, entrepreneurial behavior is the creation of services based on different types of skills. In the 1980s, the concept of skills began to receive attention. With the development of technology and the economy, skills have gradually become regarded as a resource that can provide competitiveness and productivity advantages for organizations [32]. From a historical perspective, the term "skill" has always been used to refer to personal characteristics [33], usually divided into soft skills and hard skills. Soft skills can be defined as the behavioral skills required to apply hard skills and knowledge in an organization [34]. Personal skills, interpersonal skills, and business skills are widely accepted entrepreneurship soft skills in the entrepreneurial community [35] and are a set of skills and talents of an individual [36]. The view that these three soft skill areas are key to cultivating a successful entrepreneurial mindset and skills has been increasingly accepted by experts [37]. In addition, with the intensification of competition and the continuous changes in customer taste and preferences, the modern business environment has become very unstable. Therefore, enterprises must create new products that can meet customer needs and desires to be sustainable [38]. Entrepreneurial skills enable entrepreneurs to identify customer needs and entrepreneurial opportunities [39], generate new creative ideas, and develop products or services [40]. These skills are defined as product skills in the field of entrepreneurial education.

Sustainability refers to the continuity of economic, environmental, and social development. Sustainability was elaborated into 17 Sustainable Development Goals (SDGs) and 169 specific goals. SDGs are the most widely accepted plan for achieving sustainable development today [41,42].

The goal of sustainable development has become a global challenge, and its achievement requires global cooperation and multilateral action by the economic, social, political, and environmental sectors, while also utilizing the opportunities that come with it. En-

trepreneurship and SGDs complement each other; entrepreneurs can seek economic opportunities through green innovation, political and social opportunities through reducing inequality and enhancing social cohesion, and environmental opportunities through environmental protection [43]. For example, supporting women's entrepreneurship can enhance women's empowerment, thereby enhancing entrepreneurial diversity and promoting economic growth [44], therefore enhancing SDG 5. Immigrant and refugee entrepreneurship can promote social integration, reduce their dependence on welfare and foreign aid, and stimulate domestic entrepreneurship [45], which provides clear evidence for entrepreneurship's contribution to inequality reduction and social cohesion enhancement. This is related to SDG10. In addition, green entrepreneurship and innovation have recently made new progress in the fields of agriculture, packaging, energy, and manufacturing, which may directly promote sustainable production and consumption and support SDG12 [46].

Sustainability awareness is defined as understanding the vulnerability of the environment and the importance of protecting the environment from the perspective of ecological awareness [47]. At the macro technical level, it is about the awareness of the biophysical environment, human interactions, and influences [48]. Our resources and ecosystems need to be respected and protected, and humans should take more responsible and conscious actions [49]. At the micro level, universities have begun to make significant contributions to raising public awareness about entrepreneurial education [50], and, at the same time, can motivate students to find solutions to these sustainability problems [51]. Universities should seek to promote a positive attitude towards the environment in the process of curriculum creation through their substantive functions [52] and engage in greening from different perspectives in order to achieve sustainable environmental education [53]. This can be integrated into entrepreneurial education as well.

### 2.2. Theoretical Model and Hypothesis Formulation

This paper uses constructivist theory and cognitive load theory to hypothesize the relationship between ELTs and entrepreneurial education performance, sustainability efficacy, and awareness.

The first entrepreneurship course appeared in the 1940s and was offered by Harvard University [54]. In the following half-century, entrepreneurship education (EE) began to receive more attention, and many business schools began offering one or more courses on small businesses or entrepreneurship. In the 21st century, the past 20 years have witnessed a strong rise in global entrepreneurship research [55]. Research has proven that EE is not only aimed at supporting the development of entrepreneurship as a discipline, but also at providing young potential entrepreneurs with the skills and attitudes necessary for entrepreneurship and successful business operations [56].

With the advancement of technology and the transformation of teaching methods, how to better apply electronic learning technology to entrepreneurship education has become a new research topic. Constructivist theory [57,58] believes that learning is the process of constructing meaning with the help of necessary learning resources in a certain context. Piaget pointed out that learning is a positive construction process; ELTs can provide learners with opportunities to explore and experiment with new information, leading to deeper learning [59]. Traditional entrepreneurship courses that incorporate ELTs help university graduates to find jobs in large public organizations and multinational companies or start their own businesses [60]. Cognitive load theory [61] assumes that the human cognitive structure consists of working memory and long-term memory, and that to store knowledge in long-term memory in the form of patterns is the main purpose of teaching. A schema organizes facts based on the usage of information components, providing a mechanism for knowledge organization and storage, which can lessen the workload of working memory. Research has shown that ELTs have potential value in improving student learning, as the sensory memory of images, videos, and other materials is easier to preserve in long-term memory than the memory left by simple language teaching [62]. However, not all ELTs have a positive impact on student learning; excessive transient information may lead to

cognitive load exceeding working memory limits [63]. For example, longer animations result in a large span between visual and auditory sensory memory, which greatly increases the recall of text and finally leads to the overloading of working memory [64].

### 2.2.1. Entrepreneurial Skills

ELTs are currently crucial for universities and general lifelong learning because they help instill skills including organization, teamwork, and communication through allowing students to collaborate with each other, using forms such as discussion forums and group projects, without being limited by time or location [59]. ELTs adopted in universities can deal with either synchronous or asynchronous learning which students can personally experience in their learning environment [65], which enhances student engagement and their focus on goals [66]. The flexibility of electronic learning technology provides students with the possibility of autonomous control over learning activities. The scope and depth of learning, the types of electronic devices used, and the amount of time spent on learning can all be adjusted independently [67]. They can also enhance personalized learning by providing clear content [68]. This is very helpful for cultivating students' time management skills, as they will learn to spend different amounts of time learning different types of knowledge. Ten years ago, the most common purpose of applying ELTs in education was to support communication and collaboration (through email, social platforms, group support systems, etc.) [69].

ELTs also perform well in business and entrepreneurship courses. In the increasingly fiercely competitive business environment with a shorter product life cycle, employees must possess more advanced business skills and continuously improve themselves [70]. E-learning provides learners with additional knowledge about the market, customer, product, and business activities, as well as rapid updates on new products and skills [71]. It also helps students to understand more and be more confident in professional fields like marketing [72]. Improving inventory management and strengthening marketing and communication strategies are also skills that e-learning can provide [73]. Russell (2001) listed hundreds of studies using standard methods to compare cognitive learning outcomes of courses taught online and offline using statistical tests such as final grades, paper grades, student evaluations, etc. [74]. "No significant difference" was the conclusion. Dellana, Collins, and West (2000) also support the view that online education is equally effective as face-to-face education, as they found no significant difference in the final grades of students in undergraduate management science courses [75]. In summary, the application of ELTs in entrepreneurship and business courses has a positive impact on students' learning initiatives, learning outcomes, and learning engagement [76].

Entrepreneurship students attempt to envision their new products/services based on the satisfaction of target markets and potential customer needs [77]. ELTs provide a flexible and interactive learning environment that can enhance students' understanding, collaboration, problem-solving, and critical thinking abilities, which are crucial for cultivating strong product design abilities.

Therefore, the following hypotheses are proposed:

**H1a.** *ELTs can improve students' personal skills.*

**H1b.** *ELTs can improve students' product skills.*

**H1c.** *ELTs can improve students' business skills.*

### 2.2.2. Sustainability Efficacy

In the past decade, there has been significant development in guidelines for embedding student sustainability efficacy (SSE) into education, but it is still quite complicated, with many different definitions and operations framework. Compared to developing concepts, focusing on learning outcomes is more useful because learning outcomes provide

important information on course design and interaction with students to academic staff [67]. When studying sustainability issues, students need both theoretical knowledge and the ability to discover and solve problems [78]. Given the limitations of traditional classroom lectures, Lambach proposed that the flipped classroom model may be a more suitable choice for students to pursue analysis and complex knowledge acquisition [79]. Students are supposed to develop the skills of communication, leadership, product design, and business model development and presentation. These skills can be developed with ELTs such as electronic learning forums, real-time chats, or uploading videos. E-learning can facilitate the integration of real-world examples and case studies into lessons, allowing students to see the practical application of sustainability principles. Through virtual field trips, interviews with sustainability experts, and interactive case studies, students can gain insights into how sustainability is implemented in various industries, organizations, and communities. This practical exposure helps students understand the relevance and impact of sustainability in their own lives and careers [80]. Commitment to action is important, and it is expected that "changes in values, attitudes, and behaviors" will be the result of effective sustainability education. As a result, the following hypothesis is proposed:

**H1d.** *ELT can improve students' sustainability efficacy.*

2.2.3. Entrepreneurship Skills and Sustainability Awareness

Entrepreneurship skills were identified by the United Nations in 2016 as a key factor in promoting social cohesion, reducing disparities, and opening doors for all [81]. People are gradually realizing that entrepreneurial skills can affect social development, promoting the improvement of production capacity, promoting enterprise creation, and expanding the pool of opportunities to all individuals. Entrepreneurs with excellent entrepreneurial skills have the opportunity to participate in sustainable economic development and benefit from it [82].

Entrepreneurial skills can significantly enhance students' sustainability awareness in several ways [83]. Skills like communication and collaboration can help students understand the complexity of sustainability issues and the need for flexible, innovative solutions [84]. Skills related to product development, such as product design, can help students understand how products can be designed or modified to be more sustainable [85]. This may include using renewable materials, using green energy, reducing resource consumption, or creating products that solve environmental problems. Knowledge of business areas can help students understand the need for sustainability in business [86]. They can learn how sustainable practices can lead to cost saving, meet consumer demands, and provide product advantages. These skills can help students understand how individuals and organizations can contribute to sustainability. They can learn to lead sustainability initiatives, influence others to adopt sustainable practices, and work effectively in teams to achieve sustainability goals.

The willingness of students to participate in the sustainable development agenda is related to their entrepreneurial skills; for example, students with rich entrepreneurial skills are more willing to engage in creative professions [87]. Providing students with entrepreneurial skills will help them accommodate to constantly changing conditions as well as meet the requirements of sustainable development for employees and entrepreneurs [82] In the context of the above discussion, we propose the following hypotheses:

**H2a.** *Personal skills can enhance sustainability awareness.*

**H2b.** *Product skills can enhance sustainability awareness.*

**H2c.** *Business skills can enhance sustainability awareness.*

### 2.2.4. Sustainability Efficacy and Sustainability Awareness

Researchers have shown that targeting sustainability awareness may be a way to promote an increase in sustainable behavior [88]. The theory of planned behavior (TPB) believes that all factors that may affect behavior are indirectly influenced by behavioral intention [89]. Ajzen defines intention as "a person's readiness to perform a given behavior" [90] and states that "Behavioral intention is determined by three factors: attitude towards behavior, subjective norms, and perceived behavioral control" [91], as shown in Figure 1.

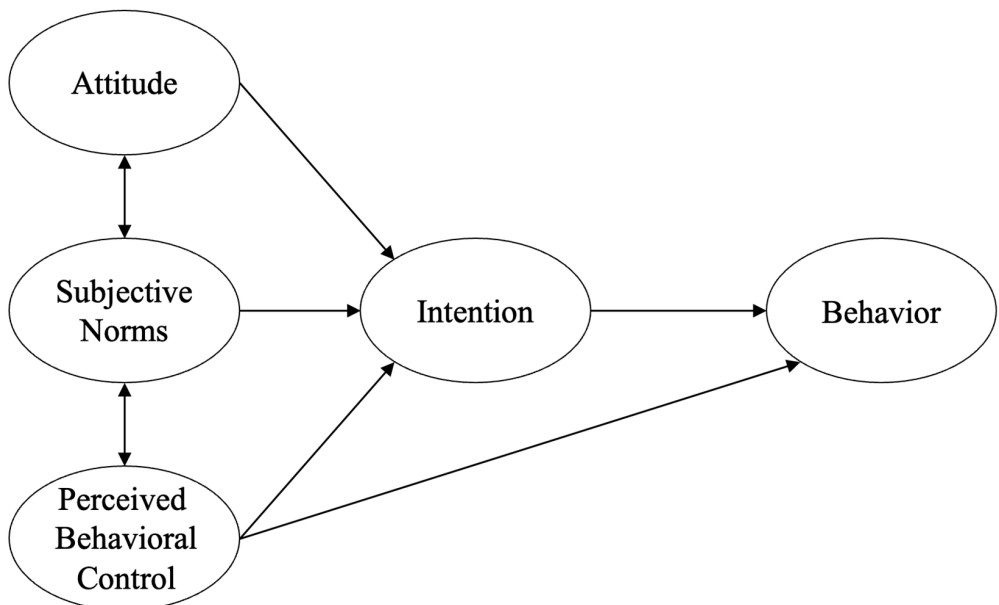

**Figure 1.** TPB model [92].

Under the premise of adopting appropriate measurement standards, there is a strong corresponding relationship between self-efficacy and behaviors [93]. Self-efficacy affects individuals' choices, short-term expenditures, and long-term persistence in activities [94–96]. People who have a lower sense of efficacy in completing a task may not make the corresponding efforts, and those who believe in their abilities will be more actively involved [97]. According to this theory, there is a positive relationship between a person's cognitive level and their sustainable development behavior [98,99]. It indicates that improving an individual's awareness of environmental protection can be achieved by improving their level of knowledge, especially environmental knowledge [100]. Therefore, we propose the following hypothesis:

**H2d.** *Sustainable efficacy can enhance sustainability awareness.*

### 2.2.5. Control Variables (CVs)

In addition to skills and efficacy, some past studies have shown that there are other factors that may affect sustainability awareness, such as gender, academic performance, and contacts with entrepreneurial role models.

Barnas and Ridwan believe that girls have a better awareness of sustainability and demonstrate greater concern for the environment than boys [101,102], while Demaidi et al. argue that boys have a better awareness of environmental and climate change than girls [103]. There are also studies that suggest that there is no difference between genders in terms of sustainability awareness [104]. These differences indicate that gender may be one of the influencing variables that needs to be included in the model. In our study, we found that some students had working experience while others did not. We support that working experience may also affect students' sustainability awareness [105,106]. In

addition, we were interested in testing whether students' academic performance measured by all courses' GPA (grade point average) achieved in the previous semester may also have an impact on sustainability awareness [107].

During data processing, we first conducted consistency testing on three control variables: gender, working experience, and academic performance. If the test results proved that these three variables had no impact on the other variables, then these three variables would not be discussed in the following steps.

Based on the above hypotheses and the three control variables (CV1, CV2, and CV3), our conceptual model is proposed below (Figure 2).

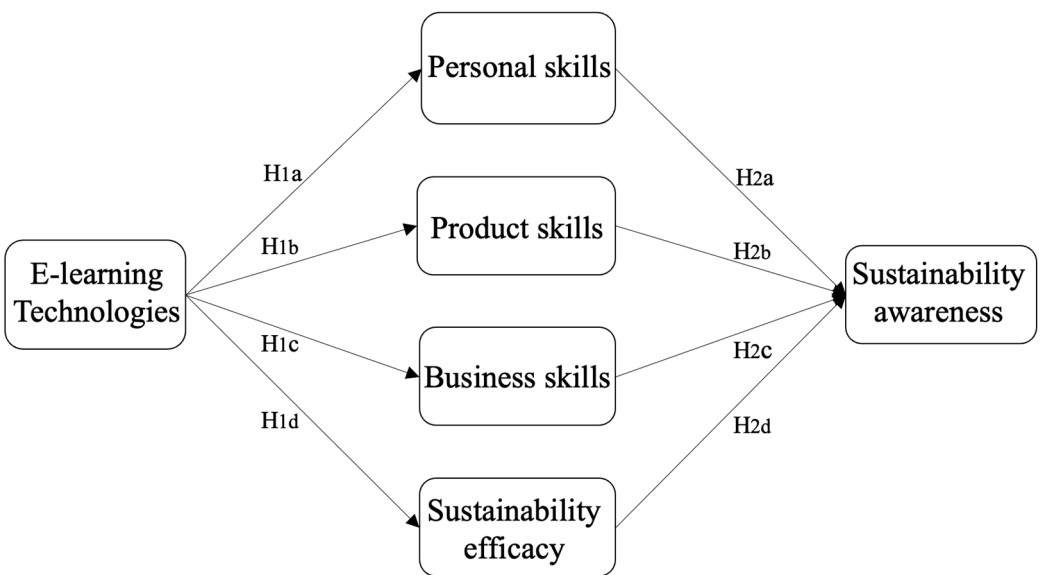

**Figure 2.** Conceptual model.

## 3. Methodology

### 3.1. Design

Most of the previous studies on teaching methods and technologies have emphasized that single online teaching is used as an alternative tool to traditional learning processes in the context of an epidemic, focusing on the exclusive use of e-learning platforms. This study uses an online questionnaire to investigate students' views on ELTs as supplementary and complementary tools to offline learning.

### 3.2. Instruments

This study proposed a 49-item questionnaire designed and distributed via Word and Canvas, covering six research areas: ELTs (9 items), personal skills (6 items), business skills (6 items), product skills (5 items), sustainability efficacy (17 items), and sustainability awareness (6 items), designed based on previous studies, as shown in Table 1.

The measure of ELTs in this study covers a learning platform called Canvas, Zoom, e-mail systems, social media, the internet as well as the latest technologies like ChatGPT, etc. A questionnaire survey was conducted to collect relevant data.

In addition, this study also designed three control variables, namely gender, GPA, and working experience, to determine whether sustainability awareness is influenced by external variables.

All responses were assessed on a 7-point Likert scale. Research has shown that compared to Likert scales with scores of 5 and below, the 7-point scale has more significant discrimination and higher reliability [108]. For online environments, a symmetrical 7-point scale interface is considered more suitable for respondents to express their evaluations, which helps alleviate their psychological burden and increase the number of respondents [109].

**Table 1.** Items in the questionnaire and references.

| | Questions | Reference |
|---|---|---|
| E-learning Technologies | I took all the lecturing as long as there is an online ZOOM link | [110–115] |
| | I reviewed lecturing recorded in ZOOM video | |
| | I interact or provide feedback with lecturer using ZOOM chat. | |
| | I use Canvas for downloading PPT and other teaching materials. | |
| | I use Canvas for submitting assignment | |
| | I frequently check E-mails for education purposes of this course | |
| | I use internet to search information about my learning in this course | |
| | I use Wechat or WhatsApp for group work and communication | |
| | I use other technologies for learning (e.g., AIchat, survey monkey, etc.) | |
| Personal skills | Communication skills | [116,117] |
| | Leadership | |
| | Responsibility | |
| | Team coordination | |
| | Team meeting | |
| | Time management | |
| Business skills | Business Plan | [116,117] |
| | Business model | |
| | Marketing plan | |
| | Financial plan | |
| | Investment | |
| | Presentation skill | |
| Product skills | New idea generation (creativity) | [116,117] |
| | New idea assessment and selection | |
| | Product design | |
| | IP search and protection | |
| | Technology road-map | |
| Sustainability efficacy | 1: No Poverty | [14,42] |
| | 2: Zero Hunger | |
| | 3: Good Health and Well-being | |
| | 4: Quality Education | |
| | 5: Gender Equality | |
| | 6: Clean Water and Sanitation | |
| | 7: Affordable and Clean Energy | |
| | 8: Decent Work and Economic Growth | |
| | 9: Industry, Innovation and Infrastructure | |
| | 10: Reduced Inequality | |
| | 11: Sustainable Cities and Communities | |
| | 12: Responsible Consumption and Production | |
| | 13: Climate Action | |
| | 14: Life Below Water | |
| | 15: Life on Land | |
| | 16: Peace and Justice Strong Institutions | |
| | 17: Partnerships to achieve the Goal | |

**Table 1.** *Cont.*

| | Questions | Reference |
|---|---|---|
| Sustainability awareness | Entrepreneurs should take care of the overall wellbeing of employees. | [101,118,119] |
| | Entrepreneurs should care about social problems at large outside the enterprise. | |
| | Entrepreneurs should make sure their operations will not pollute the environment. | |
| | Entrepreneurs should contribute to reduce pollution in the society. | |
| | Entrepreneurs should lead the company to receive continuous income and profit. | |
| | Entrepreneurs should pay attention to and to be sensitive to cash flow. | |

*3.3. Data Collection*

Data were collected from March to April 2023. Data collection took place during an innovation and entrepreneurship course at a university in Hong Kong. The course ran for one semester from January to June 2023, during which students learned about innovation, entrepreneurship, and sustainability. The class size was 101 and all students were invited to fill in the survey as part of course feedback. To avoid any common method variance, the feedback questionnaire was not marked, and only generic feedback was provided to students afterwards. The profiles of the students are shown in Table 2. This group of students mostly consisted of recent graduates and working experience was not relevant to this study.

**Table 2.** Demographics of surveyed students.

| | Frequency | Percentage% |
|---|---|---|
| **Gender** | | |
| Male | 62 | 61.4 |
| Female | 39 | 38.6 |
| **Bachelor background** | | |
| Mechanical engineering | 18 | 17.8 |
| Electronic and electrical engineering | 15 | 14.9 |
| Computer sciences and engineering | 6 | 5.9 |
| Energy engineering and related | 4 | 4.0 |
| Agricultural engineering | 7 | 6.9 |
| Bioengineering and medical-related | 4 | 4.0 |
| Ocean engineering and water-related | 5 | 5.0 |
| Chemistry-related | 5 | 5.0 |
| Physics-related | 3 | 3.0 |
| Mathematics- or statistics-related | 2 | 2.0 |
| Systems engineering or IE | 12 | 11.9 |
| Others | 20 | 19.8 |
| **Working experience** | | |
| Yes | 45 | 44.6 |
| No | 56 | 55.4 |

A total of 61.4% of the surveyed students were male and 38.6% were female. Most students had backgrounds in mechanical engineering, EE, and system engineering, accounting for 17.8%, 14.9%, and 11.9%, respectively. The other students had undergraduate backgrounds in computer science and engineering (5.9%), energy engineering and related courses (4.0%), agricultural engineering (6.9%), bioengineering and medicine-related courses (4.0%), ocean engineering and water-related courses (5.0%), chemistry-related (5.0%), physics-related (3.0%), mathematics- or statistics-related courses (2.0%), and other

fields. Among all students, 44.6% had working experience, while other 55.4% had never worked before.

*3.4. Data Analysis*

Since the model includes factors and mediating effects, this study opted for partial least squares (PLS) as the means to test its hypotheses. PLS can be the preferred solution if a problem has the following characteristics [120–122]:

1. Small sample size due to limited experimental design or survey subjects.
2. Conditions relating to independence or normal distribution that cannot be applied to SEM are not met.
3. Issues to be investigated are relatively novel and require the development of new measurement models.
4. Prediction is more important than parameter estimation.

Due to our small sample data and relatively new research topic, PLS was chosen as the research tool.

As for the three control variables, ANOVA was used to test gender, working experience, and academic performance separately. This is because GPA is not a binary variable. ANOVA can be used to analyze the differences between two or more groups of data [123].

**4. Results**

*4.1. Equality Test of Demographic Parameters*

A one-way ANOVA was used in SPSS to conduct quality testing on control variables (CV1, CV2, and CV3). The result is shown in Table 3.

**Table 3.** Equality test of demographic parameters.

|  | CV1 Gender | | CV2 Working Experience | | CV3 GPA | |
|---|---|---|---|---|---|---|
|  | F | Sig. | F | Sig. | F | Sig. |
| ELT | 0.023 | 0.881 | 0.008 | 0.928 | 0.844 | 0.723 |
| PI | 3.146 | 0.079 | 2.068 | 0.154 | 0.605 | 0.960 |
| PD | 1.306 | 0.256 | 1.306 | 0.256 | 0.990 | 0.515 |
| BS | 2.211 | 0.140 | 2.232 | 0.138 | 0.749 | 0.844 |
| ESA | 3.884 | 0.052 | 3.815 | 0.054 | 0.959 | 0.560 |
| SDG | 0.146 | 0.703 | 0.714 | 0.400 | 1.107 | 0.363 |

There is no significant relationship between the control variables and all other variables, indicating that gender, work experience, or academic performance do not have an impact on students' entrepreneurial skills, sustainability efficacy, and sustainability awareness. Therefore, in subsequent data processing, we no longer discussed the control variables CV1, CV2, and CV3.

*4.2. Validity and Reliability Tests*

For the validation of constructs, we used a threshold of 0.7 [124]. However, when it comes to the construct of e-learning, technologies like the Canvas learning platform, internet searching, e-mail, and social media did not pass the test and were deleted accordingly, since these technologies are very popular and their averages were close to 5. This does not mean that these technologies do not have impact but that there was no variance in questions about Canvas, internet searches, e-mail, and social media. In the end, three items were found to be valid and reliable in this construct, as shown in Table 3.

The sustainability efficacy construct contains 17 items but some of them were deleted as their loadings were below 0.7. This may have been because the target students majored in limited engineering fields, as shown in their background profiles in Table 2. About 51% of students were in the electronic, mechanical, and systems engineering fields and may not have felt they were able to answer some of the SDG-related questions. Loadings and

validation are shown in Table 4. The loadings in brackets are loadings after deleting items below 0.6 in ELT or below 0.7 in other constructs. The loading of one item is 0.686, which is close to 0.7 if it is rounded to the next digit. This was accepted since this construct is relatively new and few studies have tested the construct yet.

**Table 4.** Outer loadings.

| Item | Initial Loading | Final Loading |
|---|---|---|
| E-learning technologies ($\alpha$ = 0.690, AVE = 0.617, CR = 0.827) | | |
| I took all the lecturing as long as there is an online ZOOM link | 0.091 | |
| I reviewed lecturing recorded in ZOOM video | 0.760 | 0.869 |
| I interact or provide feedback with lecturer using the chat of ZOOM | 0.681 | 0.791 |
| I use Canvas for downloading PPT and other teaching materials. | −0.198 | |
| I use Canvas for submitting assignment | −0.189 | |
| I frequently check E-mails for education purposes of this course | 0.632 | |
| I use internet to search information about my learning in this course | 0.458 | |
| I use Wechat or WhatsApp for group work and communication | 0.353 | |
| I use other technologies for learning (e.g., AIchat, survey monkey, etc.) | 0.582 | 0.685 |
| Personal skills ($\alpha$ = 0.815, AVE = 0.644, CR = 0.878) | | |
| Communication skills | 0.679 | 0.726 |
| Leadership | 0.614 | |
| Responsibility | 0.799 | 0.796 |
| Team coordination | 0.854 | 0.867 |
| Team meeting | 0.813 | 0.815 |
| Time management | 0.679 | |
| Product skills ($\alpha$ = 0.878, AVE = 0.671, CR = 0.911) | | |
| New idea generation (creativity) | 0.795 | 0.818 |
| New idea assessment and selection | 0.816 | 0.811 |
| Product design | 0.783 | 0.791 |
| IP search and protection | 0.820 | 0.805 |
| Technology road-map | 0.872 | 0.868 |
| Business skills ($\alpha$ = 0.911, AVE = 0.691, CR = 0.930) | | |
| Business Plan | 0.896 | 0.897 |
| Business model | 0.873 | 0.867 |
| Marketing plan | 0.791 | 0.778 |
| Financial plan | 0.780 | 0.785 |
| Investment | 0.851 | 0.852 |
| Presentation skill | 0.792 | 0.801 |
| Sustainability efficacy ($\alpha$ = 0.895, AVE = 0.544, CR = 0.915) | | |
| 1: No Poverty | 0.726 | 0.765 |
| 2: Zero Hunger | 0.705 | 0.737 |
| 3: Good Health and Well-being | 0.744 | 0.782 |
| 4: Quality Education | 0.655 | 0.712 |
| 5: Gender Equality | 0.680 | 0.711 |
| 6: Clean Water and Sanitation | 0.673 | |
| 7: Affordable and Clean Energy | 0.593 | |
| 8: Decent Work and Economic Growth | 0.634 | |
| 9: Industry, Innovation and Infrastructure | 0.462 | |
| 10: Reduced Inequality | 0.722 | 0.758 |
| 11: Sustainable Cities and Communities | 0.726 | 0.709 |
| 12: Responsible Consumption and Production | 0.695 | |
| 13: Climate Action | 0.706 | 0.707 |
| 14: Life Below Water | 0.598 | |
| 15: Life on Land | 0.588 | |
| 16: Peace and Justice Strong Institutions | 0.775 | 0.760 |
| 17: Partnerships to achieve the Goal | 0.616 | |

**Table 4.** *Cont.*

| Item | Initial Loading | Final Loading |
|---|---|---|
| Sustainability awareness (α = 0.824, AVE = 0.652, CR = 0.882) | | |
| Entrepreneurs should take care of the overall wellbeing of employees. | 0.731 | 0.762 |
| Entrepreneurs should care about social problems at large outside the enterprise. | 0.817 | 0.815 |
| Entrepreneurs should make sure its operations will not pollute the environment. | 0.765 | 0.767 |
| Entrepreneurs should contribute to reduce pollution in the society. | 0.833 | 0.879 |
| Entrepreneurs should lead the company to receive continuous income and profit. | 0.418 | |
| Entrepreneurs should pay attention to and to be sensitive to cash flow. | 0.602 | |

To verify the adequacy of the model and the reliability of the data, we choose the following criteria: composite reliability, Cronbach's alpha, and convergent validity [125]. Composite reliability is the most commonly used method [126]. In this method, reliability values between 0.60 and 0.70 are considered "acceptable in exploratory research", and values between 0.70 and 0.90 range from "satisfactory to good". Values of 0.95 and higher are problematic as they indicate that these items are redundant, thereby reducing the effectiveness of the construct [127]. In this study, all CR values were between 0.7 and 0.9, indicating very satisfactory reliability. Cronbach's alpha values exceeding 0.7 are generally considered reliable [124]. All items except ELT had values greater than 0.7. The value of ELT is 0.690, very close to 0.7. Similarly, since this construct is relatively new, we believe that a value of 0.690 is acceptable. The AVE criterion was established by Fornell and Larker to quantify convergent validity by extracting mean variance. It is believed that when the value of AVE is greater than 0.5, the construct converges [128]. In this study, all constructs conform to convergence effectiveness.

*4.3. Model Test Results*

Partial least squares 3.3.9 (PLS 3.3.9) software was used to test the hypotheses in the conceptual model and the impact of the three control variables. The results of the model test are shown in Table 5 and Figure 3, corresponding to the conceptual model in Figure 1. According to the test results, H1a, H1b, H1c, H1d, and H2d are supported while H2a, H2b, and H2c are rejected. Significance level thresholds are set at $p \leq 0.001$ ** and $p \leq 0.05$ *, respectively.

**Table 5.** Model test results.

| Code | Hypotheses | Path Loading | T | p | Test Result |
|---|---|---|---|---|---|
| H1a | E-learning→Personal skills | 0.350 ** | 3.88 | 0.000 | Support |
| H1b | E-learning→Product skills | 0.261 ** | 2.478 | 0.013 | Support |
| H1c | E-learning→Business skills | 0.359 ** | 4.857 | 0.000 | Support |
| H1d | E-learning→Sustainability efficacy | 0.287 ** | 3.084 | 0.002 | Support |
| H2a | Personal skills→Sustainability awareness | 0.262 | 1.945 | 0.052 | Reject |
| H2b | Business skills→Sustainability awareness | −0.235 | 1.293 | 0.196 | Reject |
| H2c | Product skills→Sustainability awareness | 0.158 | 0.952 | 0.341 | Reject |
| H2d | Sustainability efficacy→Sustainability awareness | 0.225 * | 2.051 | 0.040 | Support |

* $p < 0.05$, ** $p < 0.001$.

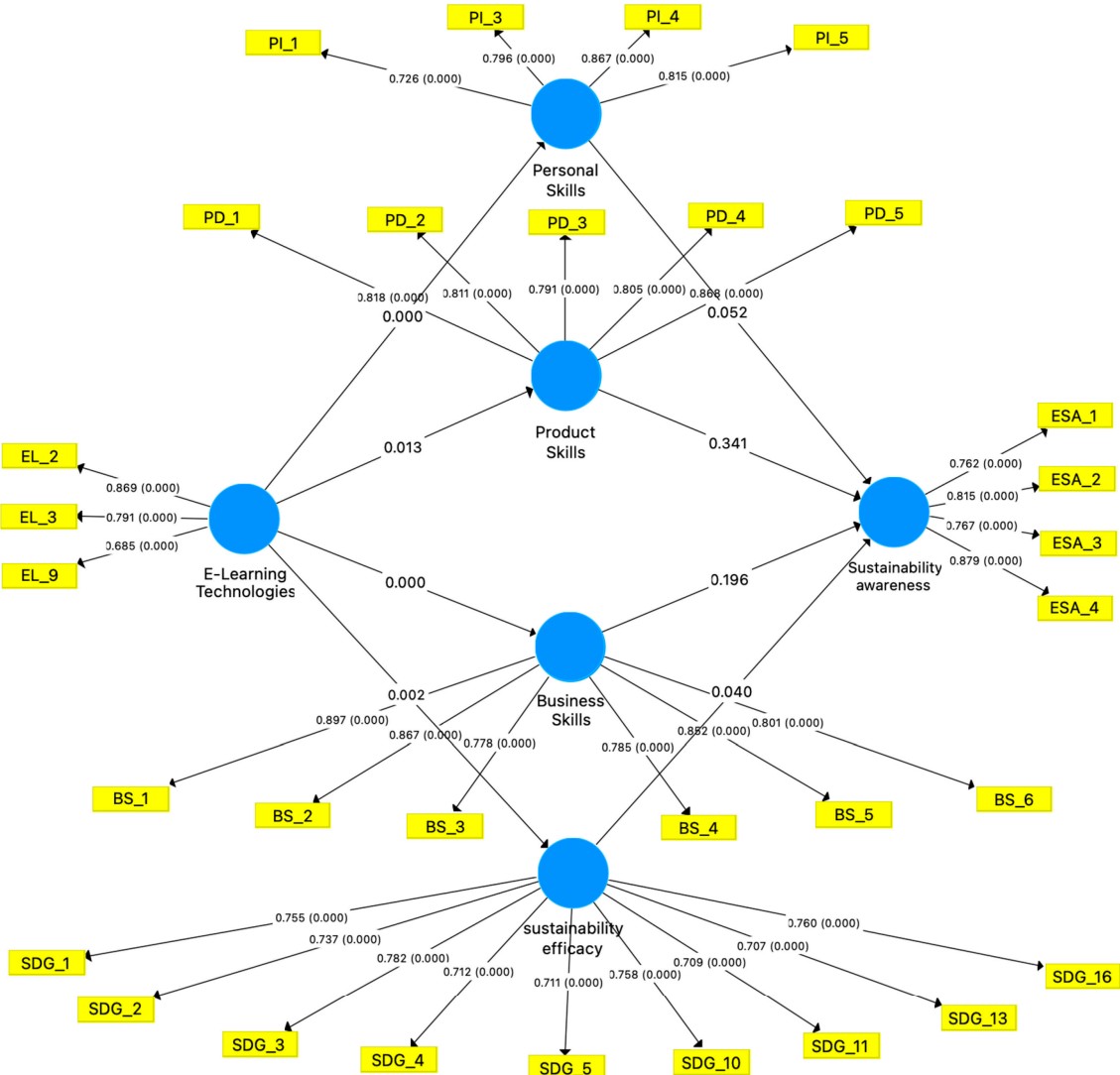

**Figure 3.** Model results.

The results imply that ELTs have a significant impact on learning performance in terms of personal skills, product skills, business skills, and sustainability efficacy. Significance levels are all equal or less than 0.001. However, only sustainability efficacy is significantly related to sustainability awareness and other entrepreneurial skills (personal, product, and business skills) do not have a significant impact on sustainability awareness. The impact of sustainability efficacy on sustainability awareness is significant but at the level of 0.04.

## 5. Discussions and Implications

This study reveals that ELTs have a significant impact on entrepreneurial skills and sustainability efficacy, but entrepreneurial skills do not enhance sustainability awareness. The implications of the results will be explored below from the entrepreneurship education and future education paradigm perspectives.

### 5.1. Impact of ELTs

This study reveals that ELTs do have a positive impact on entrepreneurial and sustainability learning performance. This has very practical implications for entrepreneurial education in the future. The International Labor Organization (ILO) proposes that entrepreneurship training must be conducted in a small class with a maximum of 30 to 50 students [129]. However, with the increasing demand for entrepreneurial education, it seems impossible

to maintain the small size of entrepreneurship classes. The class size in this case was a little bit over 100 and expected to be 160 in the coming year. There have been reports of very large classes with over 200 students in entrepreneurship courses while entrepreneurship teachers are far from enough. ELTs can be supplementary to traditional education and help universities to offer entrepreneurship courses while maintaining large class sizes.

Having said that, the impact of ELTs is different among the three entrepreneurial skills and sustainability efficacy. Both the impact on product skills and its significance level are relatively low. The impact on personal and business skills is 0.35 and 0.36, respectively, with a significance level of $p < 0.001$, while the impact on product skills is about 0.26, $p < 0.05$. That means the impact of ELTs on physical and visible things is more limited and weaker. The most likely cause of this phenomenon is the lack of experience. In Hong Kong, group assignments are an important component of higher education, where students form teams to complete assignments. Due to the characteristics of ELTs, students can organize group activities without time and location constraints [67–69], which can help them effectively exercise personal skills and business skills. However, product skills are difficult to obtain directly from text or language but require practice and experience [130]. E-learning, being primarily digital and remote, may not provide the same level of hands-on engagement. In this course, students only had access to text and image knowledge, without practical experience. Therefore, offline components such as laboratory support or the requirement for physical prototype must be included to balance the online part of learning. The impact on sustainability efficacy is also relatively lower than that on personal and business skills. This may be because it is purely personal perception instead of personal experience. It is recommended that projects related to sustainability be provided to students so that they can be involved with sustainability issues during the course of learning.

*5.2. Impact on Sustainability Awareness*

It is not very surprising that only sustainability efficacy positively influenced sustainability awareness. This is consistent with the theory of planned behavior [90–93], in which efficacy influences intention. However, the three kinds of entrepreneurial skills did not significantly affect sustainability awareness, which contradicts previous hypotheses. Previous research suggests that entrepreneurial skills can enhance students' awareness of sustainable development [83], while the results of this study indicate that attending entrepreneurial courses is not strongly correlated with sustainability awareness at the learning stage. The education system in Hong Kong places greater emphasis on academic performance and traditional subject learning, so students may focus more on improving their entrepreneurial skills by improving their test and homework scores. Therefore, their entrepreneurial skills are more related to business and innovation rather than environmental sustainability. In addition, in this course, we did not have sufficient time to collect environmental education resources, which may also be one of the reasons why students could not effectively link entrepreneurial skills with sustainability.

While entrepreneurial education can potentially enhance students' sustainability awareness, there might be several reasons why it does not always have this impact. First, not all entrepreneurial education programs emphasize sustainability. If sustainability is not integrated into the curriculum, students may not develop a strong awareness of sustainability issues. Second, many entrepreneurial programs focus on traditional business skills and objectives, such as profit maximization, rather than sustainable business practices. Third, entrepreneurship often involves a focus on short-term goals and immediate results, which may conflict with the long-term perspective required for sustainability. Fourth, without applying what they have learned in real life, students may not fully understand the relevance of sustainability to entrepreneurship. Finally, sustainability is a complex, interdisciplinary issue which may be related to engineering, technologies, and social or cultural issues. If entrepreneurial education is not integrated with other disciplines, such as environmental science or social studies [131,132], students may not develop a comprehensive understanding of sustainability. Future entrepreneurship education should address

these issues properly. It is suggested that entrepreneurship education may cover topics such as sustainable business practices, social entrepreneurship, and environmental and social issues. Students should also be encouraged to develop business ideas that contribute to sustainability, such as green products or services.

*5.3. Implications for Future Learning Paradigm*

It is widely believed that, after COVID-19, education will not be the same as before. The pandemic has indeed sped up the process of transitioning towards digital online learning or e-learning. However, education will not be 100% online or 100% offline. It will be a hybrid learning mode which combines both online and offline activities. The original hybrid mode which occurred during COVID-19 mostly refers to the mode where some students were online due to sickness and/or geographical limitations while others were offline physically in the classroom. The emergence of this mode was only a passive response to the epidemic. ELTs and the latest generative AI have brought sweeping, sudden, and uninvited changes to learning, teaching, and assessments. As a result, there will be a paradigm shift toward a technology-supported hybrid learning mode for student learning. ELTs and certain types of online learning will be part of the future hybrid education paradigm. This trend toward the new paradigm is not limited to entrepreneurship education but applies to all subjects.

## 6. Conclusions and Future Research

This study investigated the influence of ELTs on entrepreneurial and sustainability performance in a master's degree class. It was found that ELTs have significant relationships with entrepreneurial performance and sustainability efficacy. This result supports the paradigm shift from an offline mode before the pandemic to an online mode during the pandemic and then to a hybrid mode after the pandemic. Efforts are needed to fully understand and leverage ELTs in future education. This research also reveals that the impact of entrepreneurial skills on sustainability awareness is very limited. The implications of integrating entrepreneurial education with sustainability were discussed.

There are a few limitations to this study which can be explored further in future research. First, the sample of this study was from one class, which is sufficient for the power of statistical analysis but the sample size was not very big. Future research can be expanded to other classes. Second, the effectiveness of ELTs can be assessed from various perspectives such as learning outcomes or performance in the course, student engagement, completion rates, learner satisfaction, learning application, learning retention, cost-effectiveness, and technology performance. This paper only covers the impact on learning performance. All other measures are left for future research. Finally, from an entrepreneurship education perspective, this study only focuses on the influence of ELTs on entrepreneurial performance and sustainability awareness. Future research can explain attitudes, intentions, and behaviors based on the theory of planned behavior or other models considering ELTs as an influencing or moderating factor to study the differences in student performance when using and not using ELTs.

**Author Contributions:** Conceptualization, H.S.; methodology, H.S., S.L. and J.Z.; validation, S.L.; software, S.L. and J.Z.; investigation, H.S. and S.L.; resources, H.S.; writing—original draft preparation, H.S. and S.L.; writing—review and editing, H.S., J.Z. and R.X.; visualization, S.L.; supervision, H.S.; project administration, H.S.; funding acquisition, H.S. All authors have read and agreed to the published version of the manuscript.

**Funding:** The research reported in this paper was supported by the Strategic Public Policy Research Funding Scheme from the Policy Innovation and Coordination Office of the Government of the Hong Kong Special Administrative Region (S2020.A1.033.20S) and a matching fund by the City University of Hong Kong (9678240).

**Institutional Review Board Statement:** Not applicable.

**Informed Consent Statement:** Not applicable.

**Data Availability Statement:** The data presented in this study are available on request from the corresponding author.

**Conflicts of Interest:** The authors declare no conflict of interest.

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
