# Peer review of "The Impact of E-Learning Technologies on Entrepreneurial and Sustainability Performance"

_sustainability, doi:10.3390/su152115660_

Round 1

Reviewer 1 Report

Comments and Suggestions for Authors

You have clearly worked hard on this paper and the general idea is interesting. I do have a few concerns and I have also attached a copy with highlights and notes.

First, the background information reads in a very mechanical manner. Parts of the text look as though it could have been written by AI. This may be because most of the sections begin with standard definitions. Additionally, the background sections don't discuss similar research. Rather, the focus is on explanations of relevant, albeit basic concepts. 

Second, it appears the independent variables are being confused as 'control variables'. This means your study has multiple independent variables and the analysis should have been conducted with a MANOVA. 

Cronbach's Alpha, which was used in this study, is a validation test that is used to verify whether the respondents are all answering in a similar fashion. If you are trying to compare groups, you cannot use this test. It is a validation test, not a hypothesis test. For this test to be used, you would need a minimum of 100 participants per group. However, if you were not separating the participants by group (such as gender or GPA), then you could use this test, but all the test is telling you is whether the participants are responding in the same way. 

I do like the general premise of this article. However, the background sections need to be rewritten to include related research (most of the definitions can be deleted) and the results should be recalculated. 

Author Response

Dear Reviewer 1,

Thank you for taking time to review the R1 version of the paper titled "The Impact of E-Learning Technologies on Entrepreneurship and Sustainability Performance".

Best regards.

Reviewer 2 Report

Comments and Suggestions for Authors

1. In the last sentence of the abstract, the authors stated, "Implications are explored finally for entrepreneurship and sustainability 20 education". It is a general statement; however, the authors should beriefly and clearlt stated the implications in 1-2 sentences rather than proving a general statement.

2. The authors are recommended to discuss the problems and issues in the local setting where the study was conducted and elaborate the issues that motivated them to conduct such a study rather than just focusing on global and general issues and problems.

3. Methodology section has major flaws. It lacks an acceptable structure for an original paper. is should have the following headings: design, population and samples (Participants), instruments, data collection, and data analysis. Moreover, selection and sampling procedures are unclear. The authors did not specify statistically whether parametric or non-parametric tests should be done in this study. It will be identified when normal disctribution of the scores is determined. Data collection and data analysis sections are not informative enough

4. There are some language problems in teh manuscript. So, proofreading is required.

6. Some minor flaws were noticed in the reference section. The authors should double check the references to be consistent with the latest reference style recommended by the journal.

Comments on the Quality of English Language

Minor flaes were found. It should be double checked and proofread

Author Response

Dear Reviewer 2,

Thank you for taking time to review the R2 version of the paper titled "The Impact of E-Learning Technologies on Entrepreneurship and Sustainability Performance".

Best regards.

Reviewer 3 Report

Comments and Suggestions for Authors

Sometimes you write references like this [27], sometimes 26. Please choose a style and make them all the same. Volery (2000), Ribiero (2002) - quite old references for a filed which is shifting so quickly (Internet, Elearning, ICT). This comments refers mainly to the articles cited (not necessarily books, like Bandura's for example).

Line 99: The early forms of E-Learning went back to the 19th century (are you sure about this? also, what do you mean by time teachers?)

Line 478: then passively (what do you mean by this?)

Comments on the Quality of English Language

Abstract:

- investigates on the impact of ELT

- have significant relationship (a significant)

Introduction:

- Line 30: (ICT) growing (are growing)

- Implications are explored finally (finally, implications are explored...)

- Line 70: few research (little research)

- Line 73: setting questions (?)

Literature review:

0- 2.1.1. Changing of delivery modes on education (in)

Line 138: enhance SDG 5. (enhancing)

Line 152: level with 

Line 200: has been (was)

Line 305: than boy (boys)

Line 309: and need to be included (that needs)

Line 313: may also has (have) 

3. Methodology and Data Collection

I interact or provide feedback with lecturer using the chat of ZOOM (Zoom chat)

Entrepreneurs should make sure its operations (their)

Line 395: It is found there is no significant differences in term of (there are no significant differences in terms of)

Line 417: This has a very practical implications 

Line 425: while main certain large size (maintaining?)

Line 453:  in the reality, \, students (check this)

Line 467: on sustainability aware (awareness)

Conclusions:

Line 494: is sufficient in terms the power of statistical analysis (for)

Line 502: Future research can be explained to attitude (expanded?)

Author Response

Dear Reviewer 3,

Thank you for taking time to review the R3 version of the paper titled "The Impact of E-Learning Technologies on Entrepreneurship and Sustainability Performance".

Best regards.

Reviewer 4 Report

Comments and Suggestions for Authors

This is a well-executed article that addresses an interesting topic and fits well within the journal's scope.

The main issue with this article is the anecdotal and limited nature of the data collection experience ("a master class on entrepreneurship"). However, the PLS model provides a solution to this problem, and its use is well justified in the text.

All reference numbers should be included in brackets. Many of the references are entirely unnecessary, and some are quite old (a large percentage of references are over 20 years old). It is recommended to select only those that truly contribute to the study.

The literature review (and the first part of the abstract) is filled with common-sense ideas. Is it necessary to have specific sections, such as 2.1.2 or 2.1.4, to address such basic ideas so briefly?

To justify that 'Research has shown that electronic learning tools may significantly improve students' learning and memory (...),' an old research study is cited that is not openly accessible and whose results appear, a priori, rather limited. It is recommended to include a critical perspective on these aspects, emphasizing the importance of the ethical dimension when addressing entrepreneurship education.

 The included graphs help to better understand both the theoretical framework and the results.

Finally, it is recommended to enhance the discussion by further engaging with the literature referenced at the beginning.

Author Response

Dear Reviewer 4,

Thank you for taking time to review the R4 version of the paper titled "The Impact of E-Learning Technologies on Entrepreneurship and Sustainability Performance".

Best regards.

Round 2

Reviewer 1 Report

Comments and Suggestions for Authors

Enough of the paper was rewritten. While I would have conducted this study differently, I am happy with the changes. 

Author Response

Dear Reviewer,

Thank you very much for your previous suggestions, which helped me learn new theories and methods. In future paper writing, I will pay more attention to the issue of variables, choosing more suitable methods.

Best regards

Reviewer 2 Report

Comments and Suggestions for Authors

Thank you for the revisions you make in the manuscript. It is much better right now. However, it still needs some amandments. The heading" methodology and data collection" should be changed to "Methodology" only or "materials and method" based on the journals's guidelines. Then " data collection" should appear before "data analysis" section discussing how the data were collected. The process should be explained.  For the "design" section you need to discuss the type of research and its justification ( whether it is experimental study, quasi-experimental, survey, etc.). So scoring system of the questionnaire or items included in the questionnaire should be moved to another heading "instruments"

Author Response

Dear Reviewer 2,

Thank you for taking time to review the R2 version of the paper titled "The Impact of E-Learning Technologies on Entrepreneurship and Sustainability Performance". I have adjusted the structure of “Methodology” section

Best regards.
